# Alcelaphine herpesvirus 1 genes A7 and A8 regulate viral spread and are essential for malignant catarrhal fever

Françoise Myster[1], Mei-Jiao Gong[1], Justine Javaux[1], Nicolás M. Suárez[2], Gavin S. Wilkie[2], Tim Connelley[3], Alain Vanderplasschen[1], Andrew J. Davison[2], Benjamin G. Dewals[1]*

1 Immunology-Vaccinology, Department of Infectious and Parasitic Diseases, Faculty of Veterinary Medicine–FARAH, University of Liège, Liège, Belgium, 2 MRC-University of Glasgow Centre for Virus Research, Sir Michael Stoker Building, Glasgow G61 1QH, United Kingdom, 3 The Roslin Institute, Royal (Dick) School of Veterinary Studies, University of Edinburgh, Edinburgh, United Kingdom

* bgdewals@uliege.be

## Abstract

Alcelaphine herpesvirus 1 (AlHV-1) is a gammaherpesvirus that is carried asymptomatically by wildebeest. Upon cross-species transmission to other ruminants, including domestic cattle, AlHV-1 induces malignant catarrhal fever (MCF), which is a fatal lymphoproliferative disease resulting from proliferation and uncontrolled activation of latently infected CD8$^+$ T cells. Two laboratory strains of AlHV-1 are used commonly in research: C500, which is pathogenic, and WC11, which has been attenuated by long-term maintenance in cell culture. The published genome sequence of a WC11 seed stock from a German laboratory revealed the deletion of two major regions. The sequence of a WC11 seed stock used in our laboratory also bears these deletions and, in addition, the duplication of an internal sequence in the terminal region. The larger of the two deletions has resulted in the absence of gene A7 and a large portion of gene A8. These genes are positional orthologs of the Epstein-Barr virus genes encoding envelope glycoproteins gp42 and gp350, respectively, which are involved in viral propagation and switching of cell tropism. To investigate the degree to which the absence of A7 and A8 participates in WC11 attenuation, recombinant viruses lacking these individual functions were generated in C500. Using bovine nasal turbinate and embryonic lung cell lines, increased cell-free viral propagation and impaired syncytia formation were observed in the absence of A7, whereas cell-free viral spread was inhibited in the absence of A8. Therefore, A7 appears to be involved in cell-to-cell viral spread, and A8 in viral cell-free propagation. Finally, infection of rabbits with either mutant did not induce the signs of MCF or the expansion of infected CD8$^+$ T cells. These results demonstrate that A7 and A8 are both essential for regulating viral spread and suggest that AlHV-1 requires both genes to efficiently spread in vivo and reach CD8$^+$ T lymphocytes and induce MCF.

**Data Availability Statement:** All relevant data are within the manuscript and its Supporting Information files.

**Funding:** This work was supported by the following grants: BGD was supported through FSR-"credit classique" from the University of Liège (FSR-S-SS-16/20) and Incentive Grant for Scientific Research "MAGIL" from the F.R.S.-FNRS (F.4501.15). AJD is funded through Medical Research Council (MRC) Programme Grant (MC_UU_12014/3). BGD is a research associate of the F.R.S.-FNRS, Belgium. The funders had no role in study design, data collection and analysis, decision to publish, or preparation of the manuscript.

**Competing interests:** The authors have declared that no competing interests exist.

## Author summary

Gammaherpesvirus entry into immune cells can result in latent infection which is associated with viral persistence and severe lymphoproliferative diseases. Gammaherpesviruses enter target cells during primary infection via a complex machinery of envelope glycoproteins. Alcelaphine herpesvirus 1 (AlHV-1) is a gammaherpesvirus carried by wildebeests without causing any clinical sign but induces malignant catarrhal fever (MCF) upon transmission to several species of ruminants including cattle. MCF is a deadly lymphoproliferative disease developing after a prolonged incubation period. In the present study, we demonstrated that the genes A7 and A8 of AlHV-1 encode envelope glycoproteins that are orthologs of Epstein-Barr virus gp42 and gp350, which regulate cell tropism switch. Impairment of A7 or A8 expression in a pathogenic strain of AlHV-1 strongly altered viral propagation in vitro. We further showed using bovine respiratory cell lines in vitro that AlHV-1 uses A7 to mediate cell-to-cell spread whereas A8 is necessary for cell-free viral propagation. Then, infection of rabbits as an experimental model to induce MCF with recombinant viral strains demonstrated that both A7 and A8 are essential for the induction of MCF. Thus, this study highlights an essential role for gp42 and gp350 orthologs in the pathogenesis of a gammaherpesvirus-induced lymphoproliferative disease.

## Introduction

Gammaherpesvirus infection results in lifelong viral persistence in the natural host through the establishment of latency, whereas reactivation from latency is responsible for transmission from one host to another and viral persistence at the population level. Primary acute gammaherpesvirus infection is usually asymptomatic, but, under specific circumstances, latent persistence can cause the development of severe proliferative diseases and cancers [1]. In particular, the two human gammaherpesviruses Epstein-Barr virus (EBV) and Kaposi's sarcoma-associated herpesvirus (KSHV) are responsible for severe oncogenic syndromes: Burkitt's lymphoma, Hodgkin's disease, post-transplant lymphoproliferative disorder and infectious mononucleosis for EBV, and Kaposi's sarcoma, multicentric Castleman's disease and primary effusion lymphoma for KSHV [2–5]. A better understanding of the oncogenic mechanisms leading to cancer development is required for the development of effective therapies, and a comprehensive knowledge of how gammaherpesviruses enter their hosts during primary infection is key to the development of preventive measures targeting primary infection and reactivation. Although humanized mouse models are providing new perspectives into the pathogenesis of human gammaherpesviruses [6], much can also be learned from studying animal models of gammaherpesvirus infection [7,8].

Alcelaphine herpesvirus 1 (AlHV-1) is a gammaherpesvirus belonging to the genus *Macavirus* that includes various viruses involved in malignant catarrhal fever (MCF). AlHV-1 naturally infects wildebeest (*Connochaetes* sp.), and transmission is thought to occur in general during the first months of life via ocular and nasal secretions [9,10]. Importantly, most wildebeest naturally carry AlHV-1 infection, and, although the virus establishes persistent infection in this species, wildebeest do not develop any clinical sign [11]. However, upon transmission to related species, such as members of the subfamily *Bovinae* (including domesticated cattle), AlHV-1 can induce MCF, which is an acute, sporadic and fatal pan-systemic lymphoproliferative disease. The impact of MCF on local pastoralist populations has largely been underestimated, with recent reports demonstrating that MCF is a prominent cattle disease with highest economic and social impacts in regions of East-Africa subject to seasonal wildebeest

migrations [12–14]. In addition, MCF has been reported throughout the world in game farms or zoological collections in which mixed ruminant species including wildebeest are kept [15].

Recent data have demonstrated that MCF is caused by activation and proliferation of latently infected CD8$^+$ T cells [16–19], and that viral genome persistence in CD8$^+$ T lymphocytes through expression of the latency-associated nuclear antigen encoded by gene ORF73 is essential for MCF induction [18]. Interestingly, MCF is similar to the pathology caused by the New World monkey gammaherpesviruses saimiriine herpesvirus 2 and ateline herpesvirus 2 and 3, which also infect T lymphocytes and cause peripheral T cell lymphomas in tamarins, marmosets or owl monkeys [20]. Despite recent work investigating the pathogenesis of wildebeest-derived MCF in cattle and in the rabbit experimental model, the mechanisms by which AlHV-1 triggers cell proliferation and activation during latency-like infection of CD8$^+$ T cells to induce a peripheral T cell lymphoma-like disease remain unresolved [18,21–25]. In addition to the oncogenic mechanisms leading to lymphoproliferation, another important aspect of the development of preventive control strategies is the understanding of the first steps of host entry during primary AlHV-1 infection, from attachment to the mucosal barrier to infection of CD8$^+$ T lymphocytes.

Herpesviruses enter target cells using complex mechanisms involving numerous virion envelope glycoproteins. The functions of these glycoproteins range from tethering virions to target cells, to mediating envelope-membrane fusion, to delivering viral capsids into the cytoplasm [26]. Multiple envelope glycoproteins and their dynamics of expression on virions regulate the switching of cell tropism, which is important for enabling viruses to infect different cell types during host colonization and then reach the cell populations in which they establish latency [7,26]. The core fusion complex involved in entry is composed of glycoprotein B (gB) and glycoproteins H and L (gH/gL) and is shared by all herpesviruses. In addition, gammaherpesviruses express non-essential, virus-specific attachment glycoproteins such as EBV glycoprotein 350 (gp350), the KSHV K8.1 protein (K8.1), murid herpesvirus 4 glycoprotein 150 (gp150) and bovine herpesvirus 4 (BoHV-4) glycoprotein 180 (gp180) [27–32]. Moreover, EBV also express envelope glycoprotein 42 (gp42), which is involved in the switch of cell tropism and fusion in B cells [33]. Many studies have demonstrated the need for expression of and interactions between these proteins for cell entry in vitro, and animal models of gammaherpesvirus infection have provided important insights into understanding their role in host colonization [7]. However, their role in the viral life cycle in vivo in the context of lymphoproliferative disease induction is less well understood.

Studies on AlHV-1 have focused mainly on two laboratory strains, C500 and WC11, that were isolated from an ox developing MCF and a blue wildebeest, respectively [34,35]. Genome sequences of both strains are available [36–38]. Although C500 has been maintained at limited passage numbers in culture in order to retain its pathogenicity [36], WC11 has been passaged a large number of times and has been described as an attenuated strain that is no longer able to induce MCF [39]. Recent data have suggested that WC11 attenuation in cell culture is associated with the loss of parts of the genome encoding putative envelope proteins [38]. In the present study, we first confirmed the genome sequence of a stock of WC11 maintained in our laboratory, and then investigated how the loss of such genes affects viral propagation in vitro and the ability to induce MCF in vivo.

## Results

### WC11 lacks A7 and a large part of A8

The genome sequence of a German stock of attenuated strain WC11 has been published and revealed high similarity to pathogenic strain C500 despite deletions in two regions that sum to

3.3 kbp [38]. However, the duplication of a 4177 bp region that includes genes ORF50 and A6 that we had reported previously in our seed stock of WC11 [36] was not identified in the German stock. Since genomic rearrangements might have occurred independently in different viral seed stocks, we sequenced the genome of our WC11 stock. This revealed a long unique region (LUR) of 127,605 bp flanked by multiple copies of two variants of a terminal repeat (TR; 864 and 973 bp), with one copy of the former containing the duplication (corresponding to nucleotides (nt) 71303–75479). The sequence dataset of 1,300,108 sequence reads contained 333,253 reads (26%) that matched LUR and TR sequences at an average coverage depth of 608 reads/nt. Comparison with the C500 sequence [36,37] revealed a high level of similarity (Fig 1, S1 Table). In addition to the duplication, this comparison revealed several features that were also present in the German WC11 stock (Fig 1). These included a highly divergent region near the right end of LUR encoding gene A9.5 and part of gene A10, the absence of a 1175 bp sequence at the left end of LUR containing gene A1 and microRNAs 1–4 [22], and the absence of a 2174 bp sequence located centrally in LUR and containing gene A7 and the 5' region of gene A8. The duplicated sequence present in the TR also contained the same deleted sequence, indicating that the deletion had occurred prior to the duplication event [36]. Overall, the sequence of our stock of WC11 was very similar to the recently reported sequence [38], and confirmed that the genome has been subjected to several rearrangements during multiple passages in cell culture.

## AlHV-1 propagates differently in nasal fibroblasts and epithelial lung cells in vitro

The most likely transmission route of AlHV-1 to susceptible species is aerogenic [10,40]. Therefore, propagation of C500 (derived from a bacterial artificial chromosome (BAC) clone) and WC11 was compared in two cell lines selected for their respiratory origins: bovine turbinate (BT) cells and epithelial embryonic lung (EBL) cells. Titration of infectious WC11 virions (as plaque-forming units (PFU)) released into the medium and quantification of viral genomes in infected cells at 4 days post-infection (multiplicity of infection (m.o.i.) = 0.01 PFU/cell) revealed that WC11 propagated more efficiently than C500 in both cell lines (Fig 2A–2C). No significant difference in terms of infectious virions or viral genomes of WC11 was detected in BT compared to EBL cells. Conversely, significantly smaller numbers of infectious virions and viral genome copies were observed for C500 in EBL compared to BT cells. These results suggested that WC11 produces infectious virions equally efficiently in BT and EBL cells, but that C500 spreads more efficiently in BT compared to EBL cells. When cell-free infectious virion numbers were normalized to cell-associated viral genome numbers, cell-free C500 virions were significantly fewer in EBL compared to BT cells, whereas the relative numbers of cell-free WC11 virions were similar in BT and EBL cells. These results suggested that cell-free propagation of C500 is favoured in BT cells, whereas cell-associated propagation is favoured in EBL cells. Viral propagation in BT and EBL cells was monitored by measuring the percentage of fluorescent cells infected by a C500 BAC clone-derived recombinant virus expressing enhanced green fluorescent protein (eGFP). This revealed that C500 propagated more efficiently in BT compared to EBL cells. Moreover, when cell-free propagation was impaired by culturing infected cells in the presence of carboxymethylcellulose (CMC), viral propagation was significantly reduced in BT cells and remained low in EBL cells (Fig 2D and 2E). These results demonstrated that WC11 propagates more efficiently than C500 and suggest that, although BT cells enable cell-free propagation of C500, viral spread in EBL cells is limited to cell-associated growth. Finally, analysis of A7 and A8 RNA expression showed that both genes are expressed in BT and EBL cells infected with C500. As expected, no A7 RNA expression was

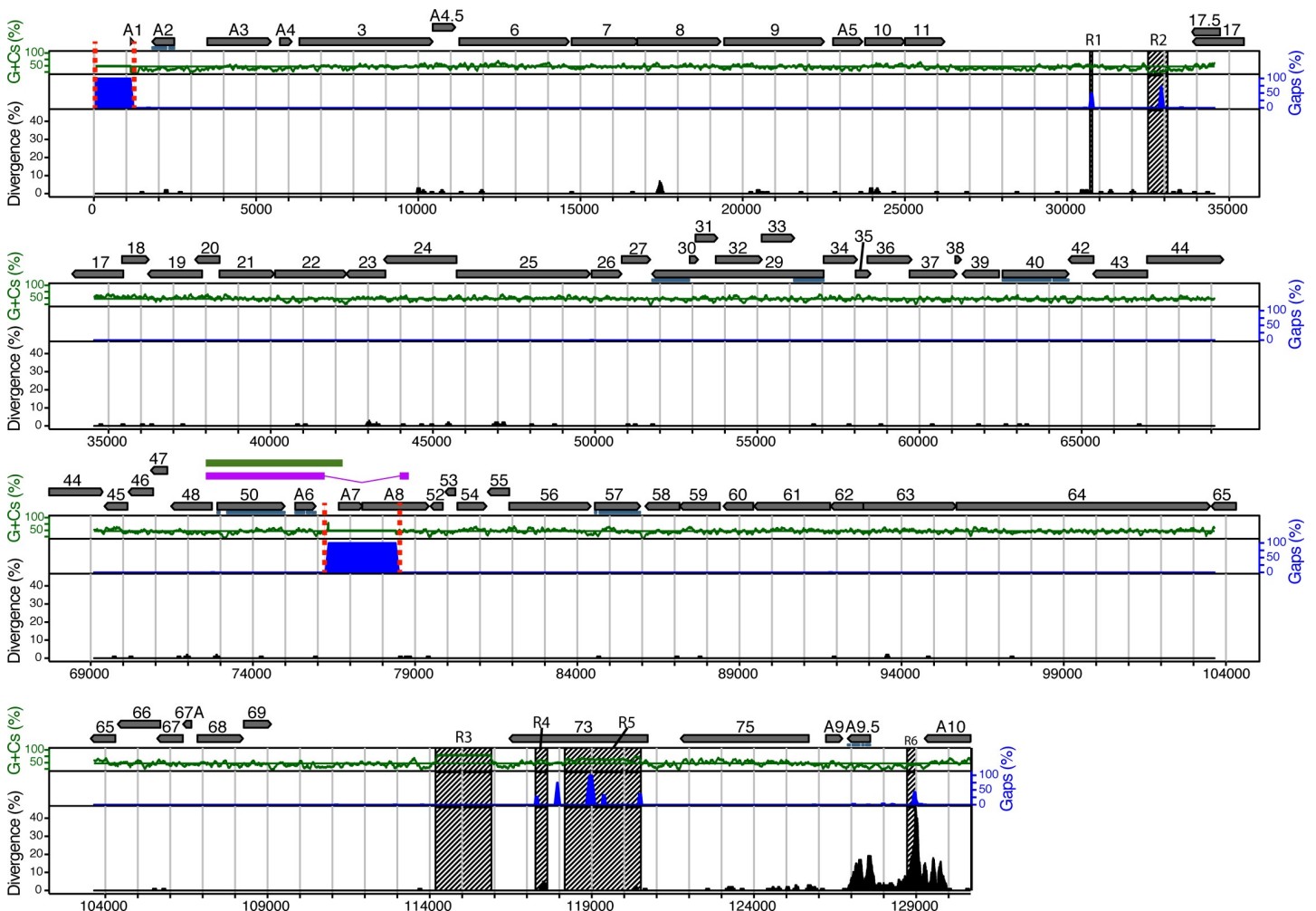

**Fig 1. Attenuated AlHV-1 strain WC11 lacks gene A7 and part of gene A8.** The illustration is based on an alignment of the LUR sequences of WC11 and C500. Genome features are represented in the upper part in relation to the C500 sequence (GenBank accession AF005370) [37]. Protein-coding regions and their orientations are depicted by grey arrows, with the ORF prefix omitted for those not prefixed by A. Exons are indicated by horizontal blue lines. Percentage divergence is shown by the black-filled curve, percentage insertions and deletions by the blue-filled curve, and percentage G+C content by the green curve, with mean G+C content shown as a horizontal green line. These percentages were measured in a 100 bp window with 3 bp increments. Tandem repeat regions (R1 to R6) are depicted as hatched areas. The magenta bar shows region duplicated in the TR of WC11. The green bar shows region duplicated in the TR of C500. Regions absent from WC11 are flanked by dotted red lines.

detected in cells infected with WC11, and expression of truncated A8 RNA remained low compared to C500 (Fig 2F).

## A7 and A8 encode homologs of the EBV gp42 and gp350 genes

The attenuation of WC11 could be due, at least in part, to the loss of A7 and A8 function. Although no roles have been attributed to these genes, they are positional orthologs of genes encoding key envelope proteins in other gammaherpesviruses [41]. A7 corresponds to EBV gene BZLF2, which encodes a type II glycoprotein containing a C-type lectin-like domain and is also referred to as gp42. A8 corresponds to EBV gene BLLF1, which encodes a highly glycosylated type I glycoprotein and is also referred to as gp350. Both gp42 and gp350 are involved in regulating the tropism of EBV for B cells [26]. Orthologs of gp42 are present in viruses in only three gammaherpesvirus genera: *Lymphocryptovirus*, *Percavirus* and *Macavirus*. However,

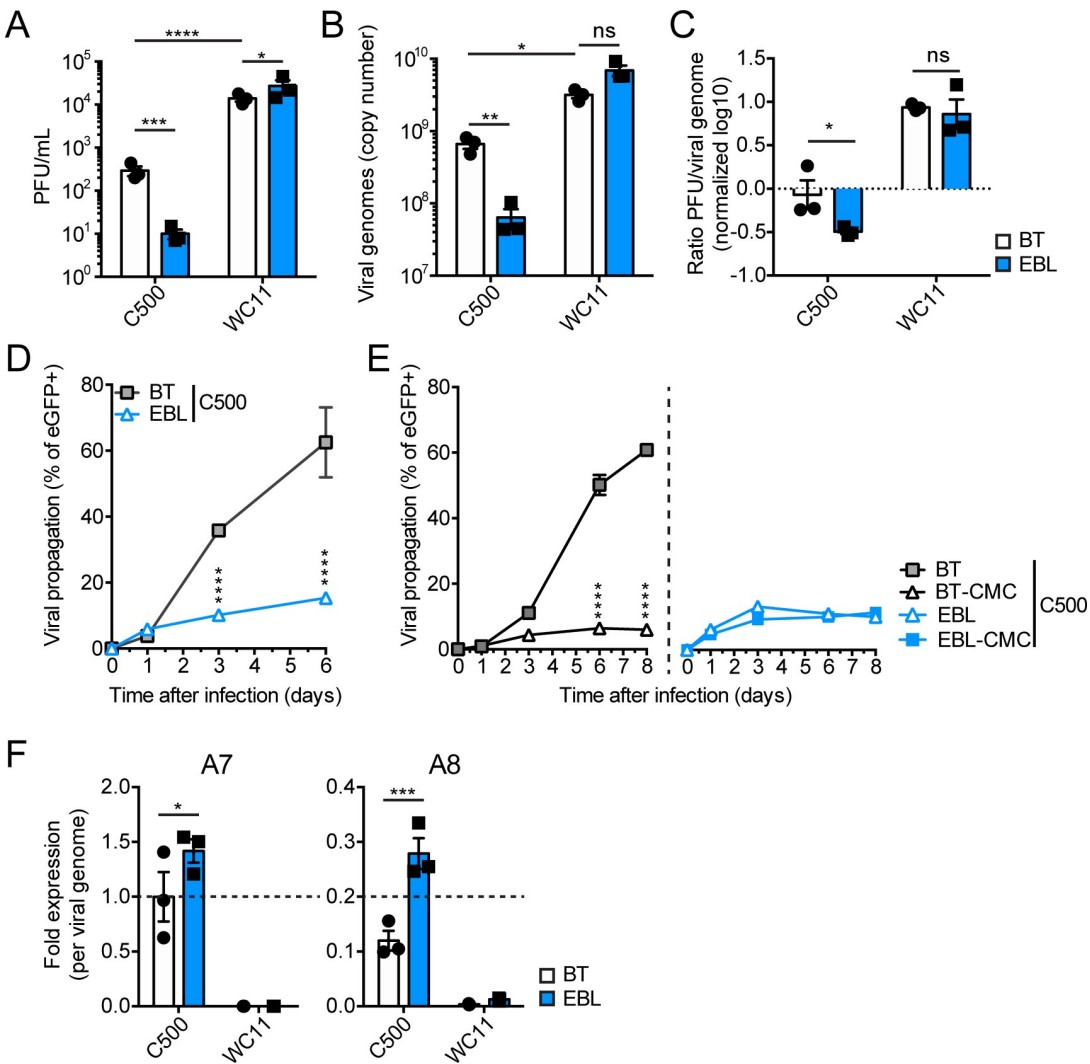

**Fig 2. Differential propagation of AlHV-1 strains C500 and WC11 in the BT and EBL cell lines.** (A) Titers of infectious virions in cell medium at day 4 post-infection (m.o.i. = 0.01 PFU/cell). Plaque assays were performed on respective cell lines (BT or EBL) to quantify viral PFUs. (B) Viral genome copy numbers derived by qPCR of DNA extracted from infected cells at day 4 post-infection (m.o.i. = 0.01 PFU/cell). Values represent the absolute copy numbers of viral genomes per well of a 12-well plate. (C) Relative ratios of infectious virions in the medium per viral genome copy in infected cells. Values were normalized on C500 in BT cells and plotted on a $log_{10}$ scale. (D) Viral propagation as percentage of eGFP$^+$ BT or EBL cells measured by flow cytometry at the given time points after infection (m.o.i. = 0.01 PFU/cell) with C500 BAC$^+$. (E) Viral propagation as in (D) with cells cultured in medium or medium containing 0.6% (w/v) CMC. (F) Quantitative PCR of A7 and A8 RNA expression at day 4 post-infection (m.o.i. = 0.01 PFU/cell). The results are shown as fold-expression values per viral genome copy normalized to C500 infection in BT cells ($2^{-\Delta Cq}$ using Cq of qPCR for ORF3 as in (B)). The results are shown as triplicates of mean ± S.D. Two-way ANOVA with Sidak's post-hoc test was used to identify significant differences ($^*P<0.05$; $^{**}P<0.01$, $^{***}P<0.001$, $^{****}P<0.0001$).

most gammaherpesviruses encode orthologs of gp350 [7]. Interestingly, the Ov8 gene encoded by the *Macavirus* ovine gammaherpesvirus 2 has recently been suggested to enhance gB/gH/gL-mediated membrane fusion [42]. In silico analyses predicted that, similar to gp42, A7 encodes a type II glycoprotein containing a C-type lectin-like domain (S1 and S2 Figs). Likewise, A8 was predicted to encode a type I glycoprotein containing numerous *O*-glycosylation sites, as reported previously for the orthologous genes in other gammaherpesviruses, such as in BoHV-4

[31,32,43]. These predictions strongly suggest that A7 and A8 are functional orthologs of EBV gp42 and gp350, respectively.

## Production of recombinant viruses containing A7 and A8 nonsense mutations

In order to investigate the role of A7 and A8 during infection, C500 viruses impaired in A7 and/or A8 expression were generated by inserting in-frame stop codons into the C500 wild type (WT) BAC clone to create nonsense mutations (S3A and S3B Fig). The A7 coding region contains several potential in-frame start codons, and we chose to target two independently at nt 39 and 207. In the C500 (WT) BAC clone, the start codon at nt 39 is present in LUR and also in the TR-associated duplication [36], whereas the start codon at nt 207 is present only in LUR. To impair A8 expression, a stop codon was inserted at nt 159 in the coding region in order to avoid disrupting the overlapping A7 coding region. To impair both A7 and A8 expression, stop codons were inserted in the A7 coding sequence using the generated A8$^{STOP}$ BAC clone. The generated BAC clones were characterized by restriction endonuclease digestion and Southern blotting before transfecting into MacT cells or MacT cells expressing Cre recombinase (MacT-Cre cells) for excising the BAC vector via flanking *loxP* sites to obtain the A7$^{STOP-39}$, A7$^{STOP-207}$, A8$^{STOP}$ and A7$^{STOP-207}$A8$^{STOP}$ viruses retaining or lacking the BAC vector (BAC$^+$ or BAC$^−$), respectively [18]. C500 BAC$^+$ and BAC$^−$ were shown to display similar fitness in vitro [44]. The generated recombinant viruses (BAC clones and BAC$^−$ viruses generated by transfection) were further verified by complete genome sequencing (S2 Table). With the exception of the nonsense mutations at the expected positions (S3C Fig), the viral genome sequences were identical. The A7$^{STOP-39}$ and A7$^{STOP-39}$A8$^{STOP}$ BAC clones displayed the insertion of the STOP codon in the LUR only and not in the TR-associated duplication [36]. Only a small part of the coding sequence of A7 (first 138 bp) is present in the duplication, which makes A7$^{STOP-207}$ the most appropriate mutant to study the role of A7.

## A7 regulates cell-associated propagation of AlHV-1 and the absence of A8 hampers cell-free propagation

The roles of A7 or A8 in viral growth in vitro were investigated by examining multistep growth kinetics and plaque size during infection of BT or EBL cells with the various generated viruses. Initial experiments showed that A7$^{STOP-39}$ and A7$^{STOP-207}$ displayed similar growth properties, and only the latter was used in subsequent experiments to avoid a risk of reversion from the duplicated sequence in the TR. In BT cells, the lack of A7 resulted in enhanced viral fitness with increased growth (Fig 3A and 3B) and production of larger plaques (Fig 3C), similar to WC11. In EBL cells, although growth of the A7$^{STOP}$ viruses was slightly increased compared to C500 (WT) (Fig 3A and 3B), A7$^{STOP-207}$ produced significantly smaller plaques (Fig 3C). WC11 displayed increased growth and produced larger plaques in both cell lines. Absence of A8 resulted in severely impaired viral growth in both BT and EBL cells, with reduced growth (Fig 3A and 3B) and absence of detectable infectivity released into the medium of infected BT cells (Fig 3B). Although the lack of A8 resulted in significantly smaller plaques in BT cells, infection of EBL cells with A8$^{STOP}$ did not significantly affect plaque size at day 6 post-infection (Fig 3C). Interestingly, absence of both A7 and A8 expression resulted in viral growth similar to strain C500 in BT cells while more similar to A7$^{STOP}$ in EBL cells (Fig 3B). Moreover, infection of BT cells with A7$^{STOP}$A8$^{STOP}$ did not significantly affect plaque size over time, but absence of both A7 and A8 generated significantly smaller plaques in EBL cells (Fig 3C). Viral propagation in the absence of A7 and/or A8 in BT and EBL cells was further assessed during infection by fluorescence microscopy in order to visualize infectious centers at day 1, 3, 4 and

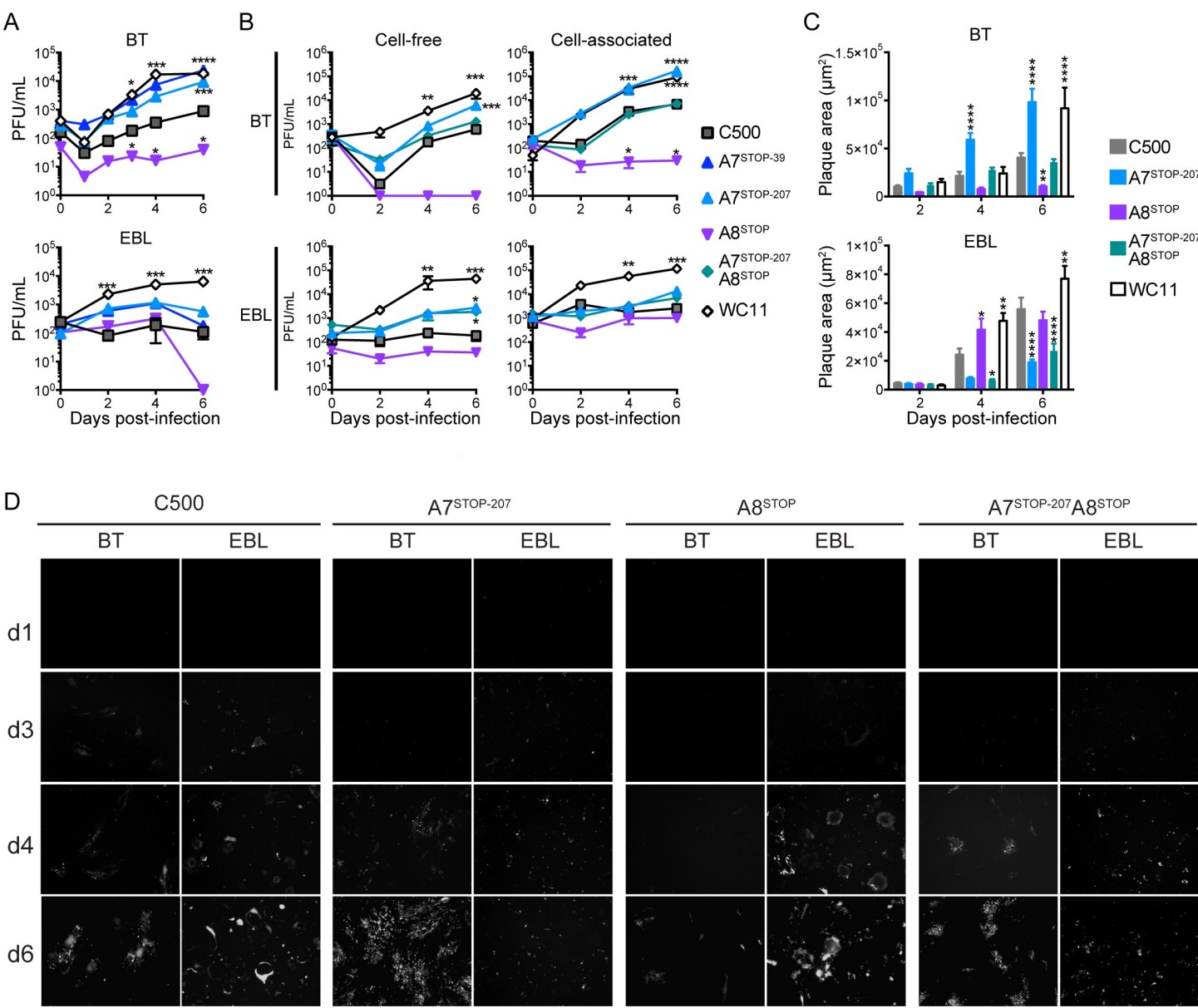

**Fig 3. A7 and A8 expression regulate viral propagation.** (A and B) Multistep growth curves of C500 BAC$^+$ WT, BAC$^+$ A7$^{STOP-39}$, BAC$^+$ A7$^{STOP-207}$, BAC$^+$ A8$^{STOP-159}$, BAC$^+$ A7$^{STOP-207}$A8$^{STOP-159}$ and WC11 in BT (top) or EBL (bottom) cells. Data are displayed as mean ± SD of measurements in triplicates. (A) Titers of cell-released and cell-associated infectious virions during infection. Plaque assays were performed on MDBK cells to quantify viral PFUs. (B) Titers of cell-released (left) and cell-associated infectious virions during infection. Plaque assays were performed on respective cell lines (BT or EBL) to quantify viral PFUs. (C) Infectious plaque areas during infection of BT (top) or EBL (bottom) cells. Data presented are mean ± SD of measurements in triplicates. (D) Microscopic analysis of viral propagation after infection of BT or EBL cells. Representative images are displayed for each time point after infection with C500 BAC$^+$ (WT), BAC$^+$ A7$^{STOP-207}$, BAC$^+$ A8$^{STOP-159}$ or BAC$^+$ A7$^{STOP-207}$A8$^{STOP-159}$ (m.o.i. = 0.01 PFU/cell). Infectivity was revealed by eGFP expression (white signals). Original magnification = 40×.

6 post-infection (Fig 3D). Together, these analyses revealed that A7 expression is necessary for viral fitness in EBL whereas A8 expression affects viral propagation in BT cells.

## A7 expression is necessary for syncytia formation and regulates infectivity

To obtain a fuller understanding of the role of A7 and A8 in plaque formation, BT or EBL cells were infected with C500 (WT), A7$^{STOP-207}$, A8$^{STOP}$, A7$^{STOP-207}$A8$^{STOP}$ or WC11 and

immunostained with a monoclonal antibody against the viral gp115 complex at day 4 post-infection (Fig 4) [45]. Fluorescent microscopy was then used to detect eGFP expression, anti-gp115 staining and DNA staining with 4′,6-diamidino-2-phenylindole (DAPI). C500 (WT) and A8$^{STOP}$ induced giant multinuclear syncytia in both BT and EBL cells, suggesting that A8 is dispensable for syncytia formation. However, A7$^{STOP-207}$, A7$^{STOP-207}$A8$^{STOP}$ and WC11 did not generate syncytia but caused a cytopathic effect with plaques formed of separated infected cells. These results suggest that A7 may be necessary for cell-to-cell viral transmission and promotion of membrane fusion.

Since propagation of C500 (WT) was mainly cell-associated in EBL cells, whereas cell-free virion release occurred in BT cells (Fig 2D), an infectivity assay was performed to investigate the requirement for contact duration between the virion envelope and the cell membrane during viral entry (Fig 5A). The efficiency of C500 (WT) in entering EBL cells was more than 10 times higher than that of entering BT cells. However, in the absence of A7, the efficiency was significantly reduced in EBL cells but increased in BT cells (Fig 5B). Thus, whereas A7 was required for effective entry into EBL cells, its absence improved entry into BT cells. The role of A8 in entry could not be investigated owing to the very low titers obtained for A8$^{STOP}$. To investigate further the role of A7 in cell-associated viral propagation, an assay using A7$^{STOP-207}$ was performed as described in Fig 2D. Inhibition of infectious virion release into the medium by CMC-impaired propagation of A7$^{STOP-207}$ (Fig 5C), thus further demonstrating that the increased growth of A7$^{STOP-207}$ is mainly due to increased cell-free propagation and virion release in BT cells.

## Impaired expression of A7 or A8 renders AlHV-1 unable to induce MCF in rabbits

To investigate the role of A7 and A8 in the induction of MCF, the experimental rabbit model was used [16,17,19,21]. Owing to the impaired growth of A8$^{STOP}$ and our consequent inability to obtain sufficient amounts of cell-free virus, rabbits were infected with BT cells exhibiting >90% cytopathic effect. Rabbits were administered intravenously with mock-infected cells or cells infected with C500 (WT), A7$^{STOP-207}$, A8$^{STOP}$ or WC11 (Figs 6A–6C, 7A and 8). Typical lesions of MCF, such as persistent hyperthermia, splenomegaly and lymphadenopathy, were observed only in rabbits infected with C500 (WT) (Fig 6A–6C, S4 Fig). Similarly, expansion of CD8$^{+}$ T cells in peripheral blood mononuclear cells (PBMCs) during infection or in the spleen and popliteal lymph nodes, and other characteristic lesions, were observed only after infection with C500 (WT) (Fig 7 and Fig 8A and 8B). While we observed viral genomes increasing over time in PBMCs of rabbits infected with the WT virus, no viral copy could be detected in any time point after infection with A7$^{STOP-207}$, A8$^{STOP}$ or WC11 viruses (Fig 8C). However, viral genomes could be detected in the spleen, liver and lung tissues of rabbits infected with the A7$^{STOP-207}$, A8$^{STOP}$ or WC11 viruses at 42 days post-infection, although at significantly lower levels than in the WT group (Fig 8D). These results suggested that the lack of A7 or A8 expression impaired the development of MCF but did not completely impaired the infectivity of AlHV-1 in vivo. Although the viral inocula were not normalized in this experiment, these results demonstrated that A7 and A8 are required to induce MCF after intravenous infection using infected cells. Total anti-AlHV-1 antibodies and virus-neutralizing antibodies were detected in all groups (S5 Fig), but the antibody response was slightly lower in rabbits infected with A8$^{STOP}$. This could have been due to the use of a lower infectious dose with this virus. Given this limitation, a further experiment was performed in which rabbits were administered intranasally with virions of C500 (WT) ($5 \times 10^{4}$ and $2 \times 10^{5}$ PFU/rabbit) or A7$^{STOP-207}$ ($5 \times 10^{4}$, $2 \times 10^{5}$ and $8 \times 10^{5}$ PFU/rabbit) (Fig 6D–6F and Fig 9A and 9B). Clinical signs of MCF were

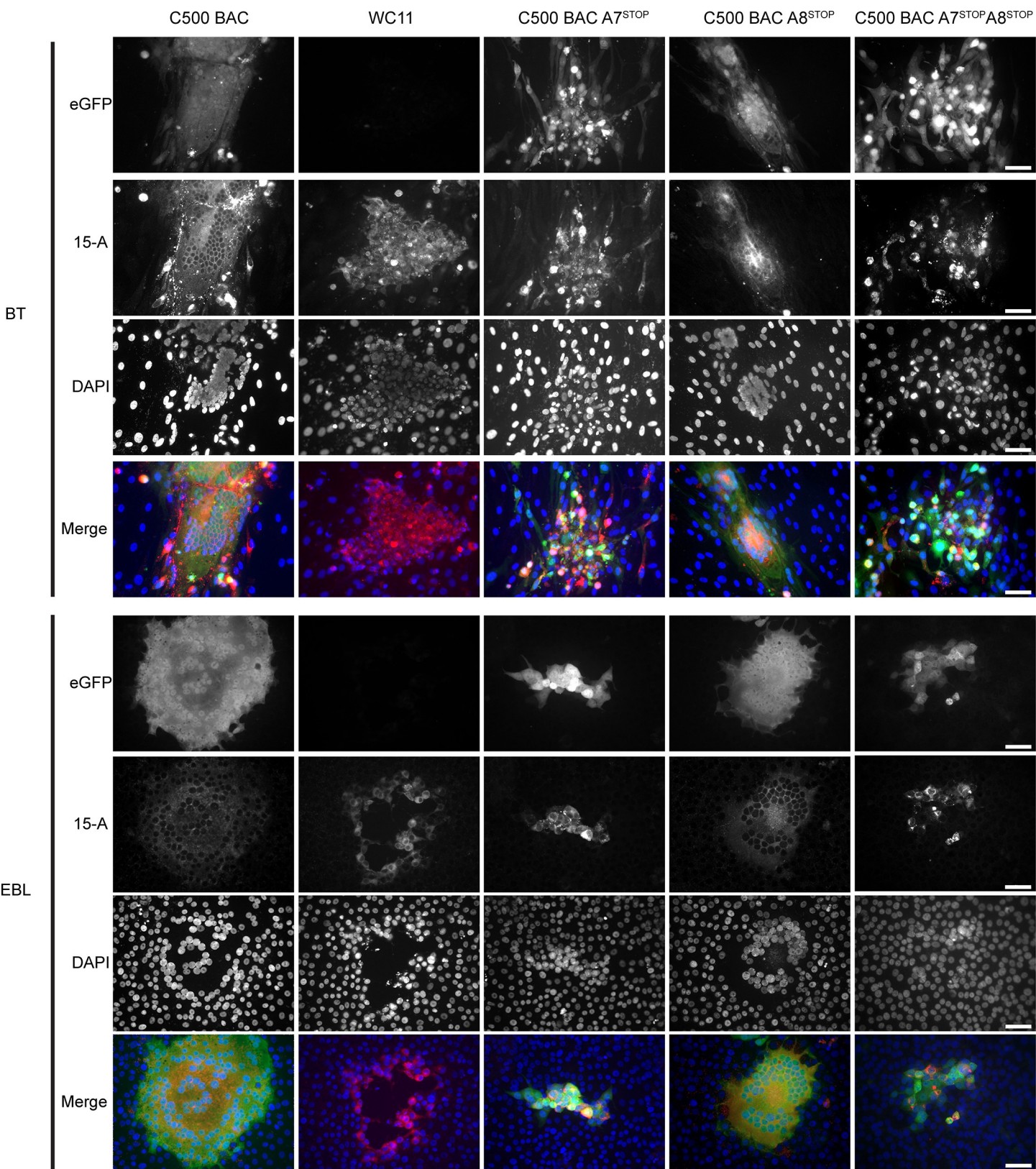

**Fig 4. Impaired syncytia formation in the absence of A7 expression.** BT or EBL cells grown on glass coverslips were inoculated with C500 BAC⁺ WT, BAC⁺ A7$^{STOP-207}$, BAC⁺ A8$^{STOP-159}$, BAC⁺ A7$^{STOP-207}$A8$^{STOP-159}$ or WC11 in presence of 0.6% CMC. At day 4 post-infection, cells were stained for glycoprotein complex gp115 (mAb 15-A; red in Merge) and DNA (DAPI; blue in Merge). Representative images are shown. Scale bars = 50 μm.

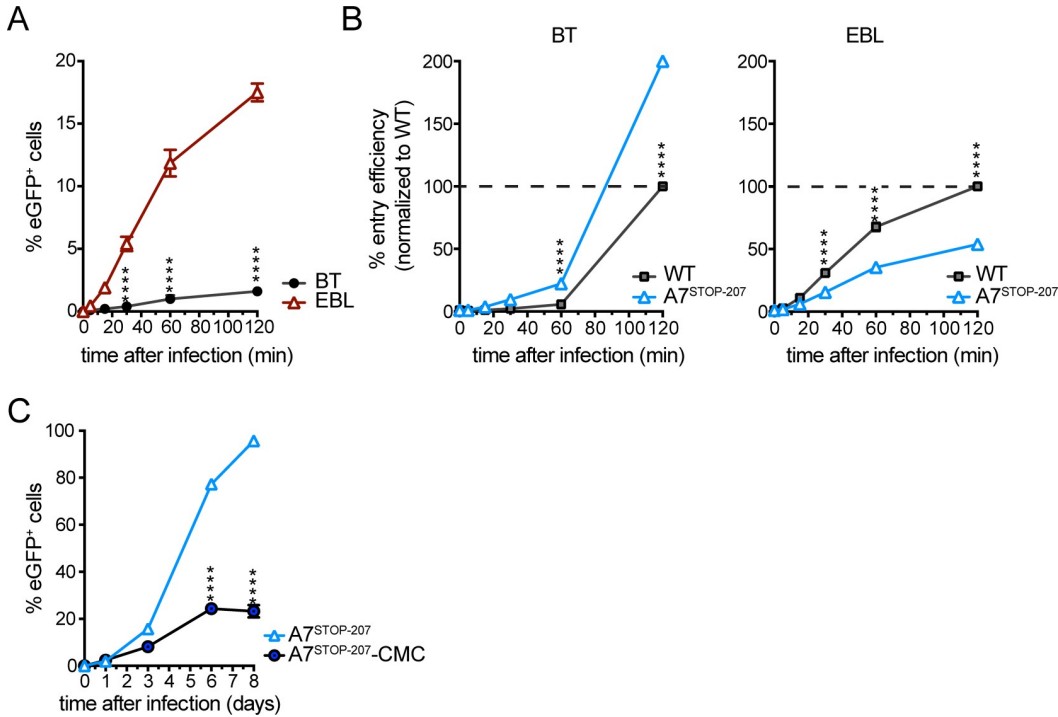

**Fig 5. Infectivity assay.** (A) BT and EBL cells were inoculated with eGFP-expressing C500 BAC+ WT (m.o.i. = 0.05 PFU/cell) for the indicated times and then washed with PBS and cultured overnight. Viral infection was assayed 20 h later by flow cytometry for eGFP expression. (B) BT and EBL cells were inoculated with eGFP-expressing C500 BAC+ WT or BAC+ A7STOP-207 (m.o.i. = 0.05 PFU/cell) for the indicated times and then washed with PBS and cultured overnight as in (A). Plotted data represent entry efficiency as the normalized percentage of C500 (WT) virus after 120 min incubation. (C) Viral propagation of BAC+ A7STOP-207 (m.o.i. = 0.01 PFU/cell) in BT cells cultured in medium or medium containing 0.6% (w/v) CMC). Data are displayed as mean ± SEM of measurements in triplicates. Two-way ANOVA with Sidak's post-hoc test was used to identify significant differences (****$P$<0.0001).

induced by both doses of C500 (WT) but not by any dose of A7STOP-207. In addition, while we observed viral genomes increasing over time in PBMCs of rabbits infected with the WT virus, no viral copy could be detected in any time point after infection with any of the infectious doses of A7STOP-207 virus (Fig 9C and 9D). Antibody response analysis revealed the presence of neutralizing antibodies in all groups (S5 Fig), with a significantly enhanced response in rabbits infected with A7STOP-207. Thus, A7 and A8 are both indispensable for the induction of MCF in rabbits.

## Ex vivo infection of CD8+ T lymphocytes in absence of A7 and/or A8 expression

We further investigated the ability of AlHV-1 to directly infect bovine CD8+ T cells ex vivo in absence of A7 and/or A8 expression. Because viral titers remain low in cell culture, a co-culture assay was used where BT or EBL cells were infected with BAC+ C500 (WT), A7STOP-207, A8STOP or A7STOP-207A8STOP viruses at 24h before the addition of 2×10^6 bovine PBMCs or *Theileria parva*-immortalized CD8+CD4− T cells (CD8TpM). Because A8STOP titers were limited, we used an m.o.i. of 0.01 PFU/cell for all virus strains (Fig 10A), and an m.o.i. of 0.1 PFU/cell for BAC+ C500 (WT), A7STOP-207, or A7STOP-207A8STOP viruses (Fig 10B). After 48h co-culture, the percentages of eGFP+ cells were measured in CD8+ T cells by flow cytometry (Fig 10 and S6 Fig). At both m.o.i., A7STOP-207 virus could infect CD8+ T cell in all conditions and

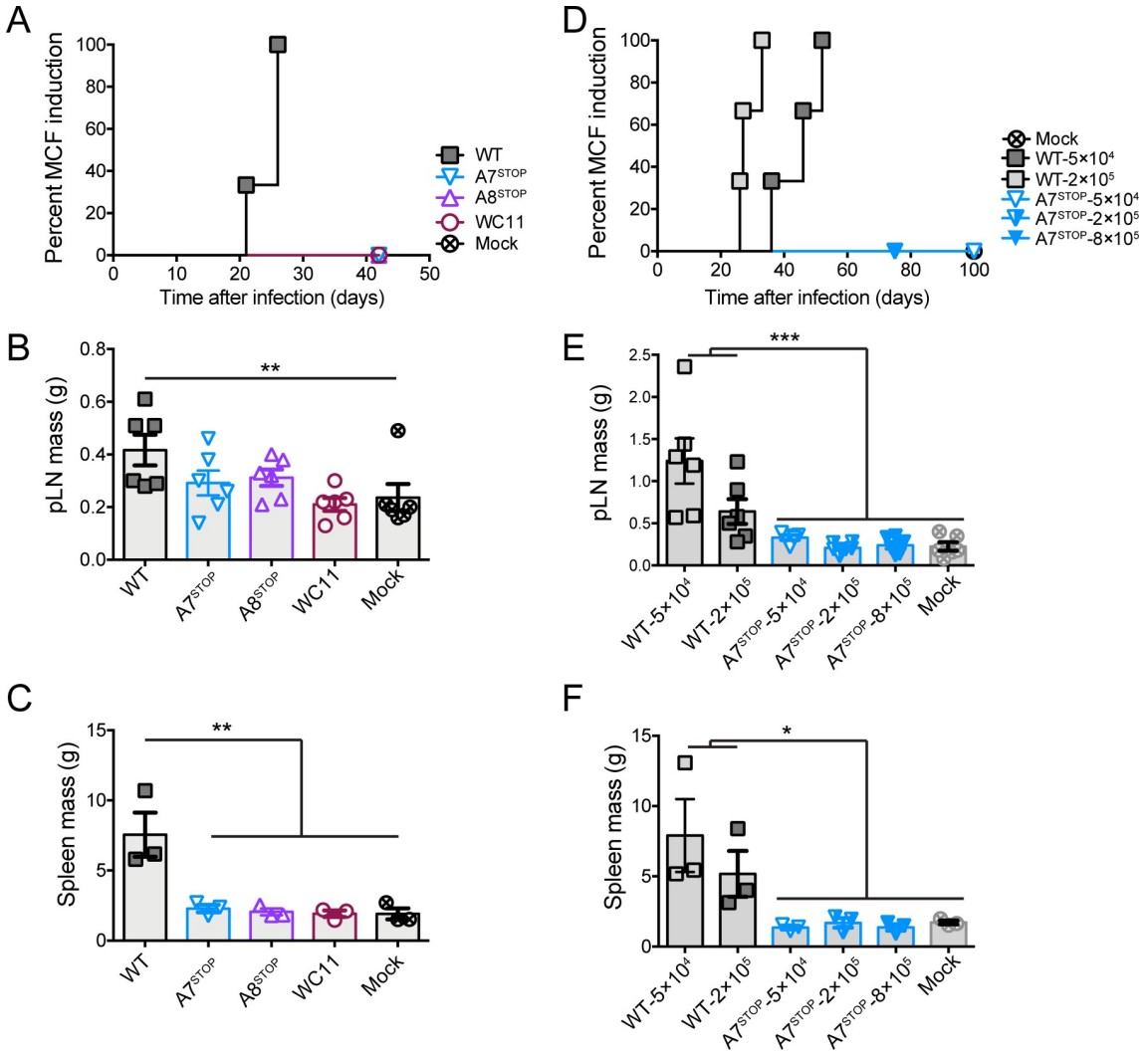

**Fig 6. Impaired expression of A7 or A8 renders AlHV-1 unable to induce MCF.** (A-C) Rabbits were infected by intravenous inoculation with 50 cm² of mock-infected BT cells or cells infected with C500 (WT), A7$^{STOP-207}$, A8$^{STOP-159}$ or WC11. (D-F) Rabbits were infected by intranasal inoculation of different doses ($5 \times 10^4$, $2 \times 10^5$ or $8 \times 10^5$ PFU/rabbit) of C500 (WT) and A7$^{STOP-207}$. (A and D) Cumulative incidence of MCF development. Popliteal lymph nodes (pLN) (B and E) and spleen (C and F) tissue mass were measured at the time of euthanasia. Bars represent mean ± SEM with data plotted for each individual rabbit. One-way ANOVA with Sidak's post-hoc test was used to identify significant differences (*$P < 0.05$, ** $P < 0.01$).

displayed higher percentages compared to C500 BAC WT virus at high m.o.i., probably due to its increased cell-free propagation observed in Fig 3 (Fig 10A and 10B). At low m.o.i, eGFP⁺CD8⁺ T cells could be detected at higher proportions after A8$^{STOP}$ infection compared to the mock control, and A7$^{STOP}$A8$^{STOP}$ virus could also infect CD8⁺ T cells almost as efficiently as the C500 WT virus during co-culture in EBL cells while the proportions of eGFP⁺CD8⁺ T cells remained similar to A8$^{STOP}$ virus in BT co-cultures (Fig 10A and 10B). While these results suggested that AlHV-1 is able to enter CD8⁺ T cells in absence of A7 or A8 expression, the differential viral growth of the mutant strains does not allow a robust comparison in their intrinsic potential of infectivity. Thus, we repeated the co-culture assay by co-culturing CD8TpM with EBL cells infected with BAC⁺ C500 (WT) or A7$^{STOP-207}$A8$^{STOP}$ viruses displaying similar growth after 4 days infection (Fig 10C). Similar proportions of eGFP⁺CD8⁺

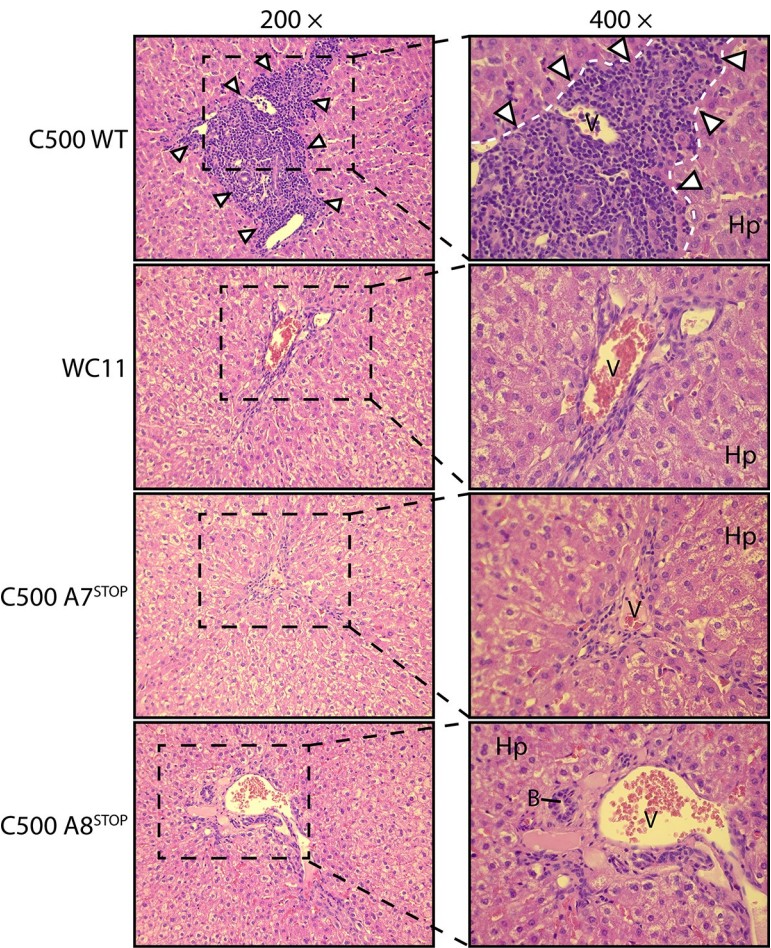

**Fig 7. Histopathological characterization of MCF lesions.** Liver sections of one rabbit representative of each group are shown. White arrowheads indicate typical infiltrations of lymphoblastoid cells. Abbreviations: B, small bile ducts; Hp, hepatocytes; V, portal veins, Equivalent results were obtained in two independent experiments. Original magnifications are indicated.

T cells after 48h of co-culture were observed (Fig 10C). Thus, a lack of A7 or A8 does not significantly impair infection of CD8⁺ T cells in co-culture conditions.

## Discussion

The sequence of our WC11 seed stock is very similar to that of a German stock reported previously [38], with the addition of a region in LUR that is duplicated in a copy of TR [43]. It is also largely similar to the sequence of C500 [35, 36]. This suggests that the attenuation of WC11 may be due, at least in part, to the region of high divergence near the right end of LUR or to one or both of the two recognized deletions, including A1 gene, a cluster of 4 microRNAs [22], A7 and A8. Notwithstanding a potential role of A1, microRNAs 1–4 or divergent sequence in A9.5 and A10, we focused here on the observation that WC11 cannot express functional A7 or A8 proteins, which are orthologs of EBV envelope glycoproteins gp42 and gp350, respectively. Regulation of gp42 and gp350 expression is essential in the EBV lifecycle because these proteins are involved, together with gB and gH/gL, in viral tropism for B cells [26]. In silico analyses showed that the A7 and A8 proteins share important sequence and predicted structural similarities with gp42 and gp350, respectively, thus suggesting that these

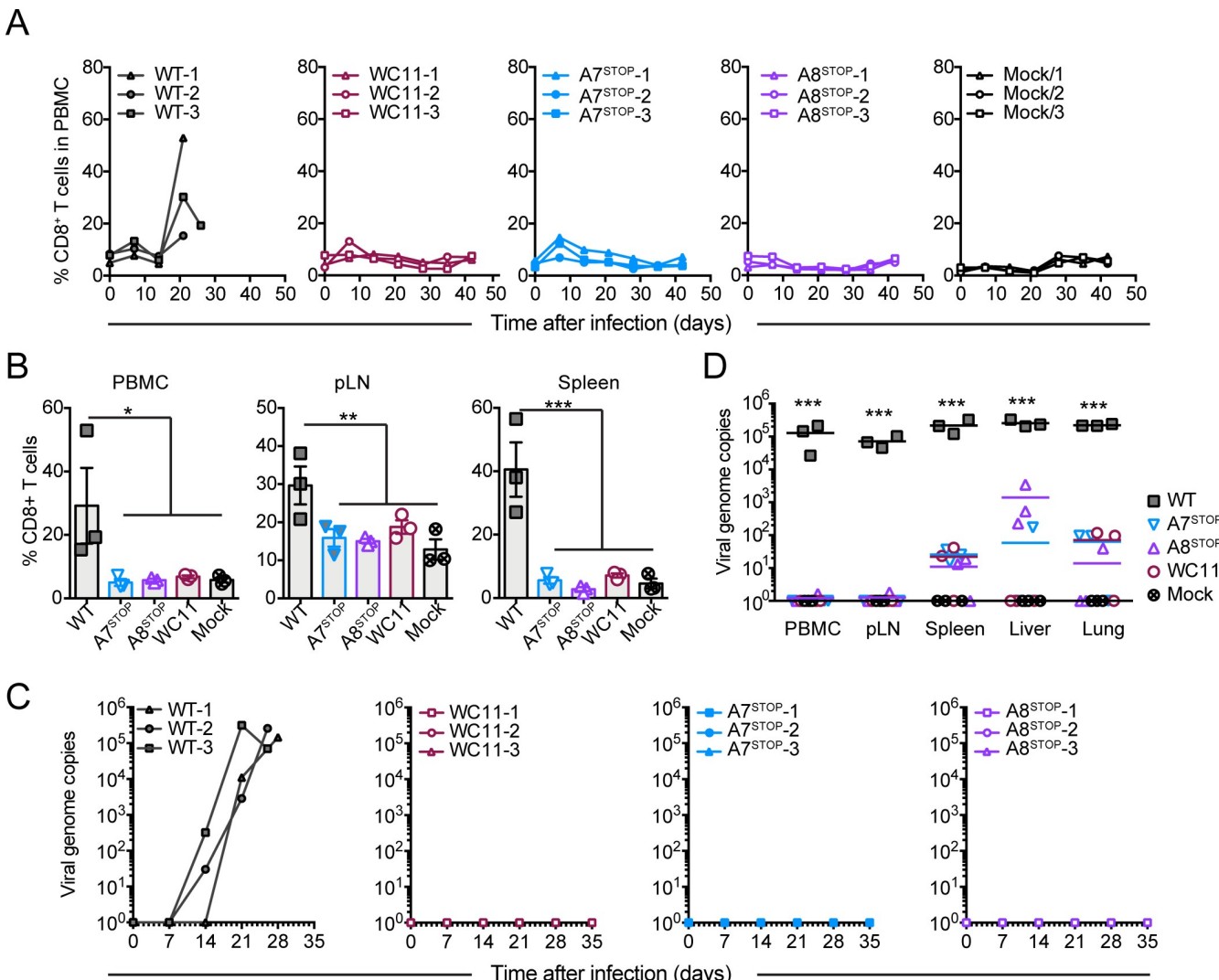

**Fig 8. Lack of A7 or A8 expression results in the absence of expansion of CD8+ T lymphocytes but persistence of viral infection after intravenous infection.** Rabbits were infected by intravenous inoculation with 50 cm² of mock-infected BT cells or cells infected with C500 BAC⁻ WT, BAC⁻ A7^STOP-207, BAC⁻ A8^STOP-159 or WC11. (A) Percentages of CD8+ T cells in PBMCs analyzed by flow cytometry at regular intervals throughout the experiment. Data are plotted for individual rabbits. (B) Percentages of CD8+ T cells in PBMCs, popliteal lymph nodes (pLN) and spleen at the time of euthanasia, as analyzed by flow cytometry. Data show mean ± SEM with data plotted for each individual rabbit. (C) qPCR of viral genome copies in PBMCs over time. (D) qPCR of viral genome copies in PBMCs, pLN, spleen, liver and lung tissues at the time of euthanasia. Real-time PCR quantification was normalized on 10⁵ copies of the cellular β-globin gene sequence. Data are plotted as individual measurements. Bars show mean values. One-way ANOVA with Bonferroni's post-hoc test was used to identify significant differences (*$P<0.05$, **$P<0.01$, ***$P<0.001$).

proteins are likely expressed in virions. Indeed, the A8 protein has been detected in purified C500 virions by proteomic analysis [46]. EBV gp350 and its orthologs in other gammaherpesviruses are major envelope glycoproteins that are highly expressed in the virion envelope [31,47,48]. Therefore, it is possible that A8 is also abundant in AlHV-1 virions explaining its detection by proteomics analyses whereas A7 could not be detected.

WC11 has been described as having lost its potential to induce MCF due to multiple passaging in cell culture [39], and such attenuation was associated in the present study with more efficient cell-free propagation in vitro. Similarly, high-passage stocks of C500 have been reported to have lost pathogenicity with enhanced cell-free virion production in vitro [49,50].

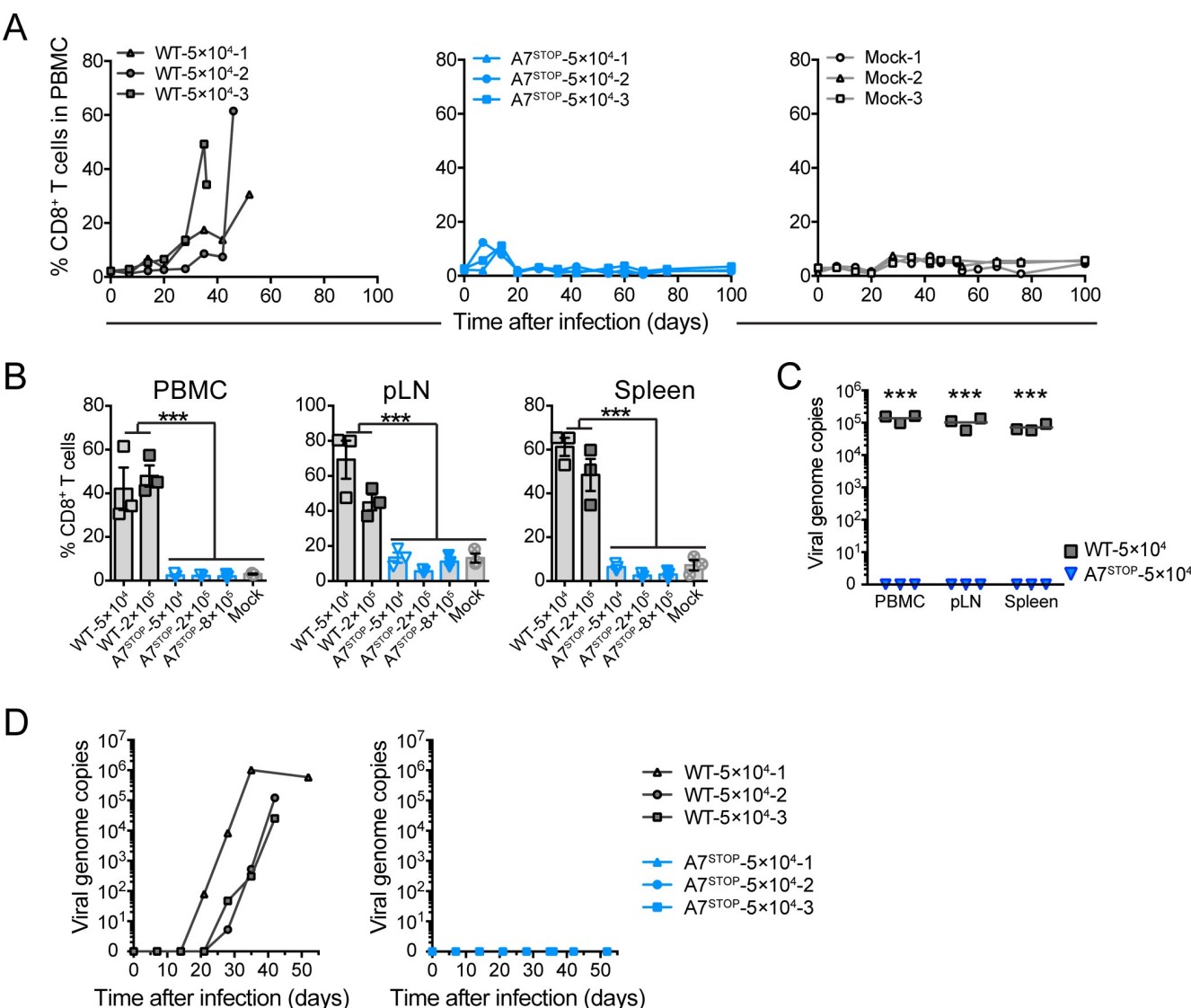

**Fig 9. Lack of A7 results in the absence of expansion of CD8+ T lymphocytes and viral persistence in peripheral blood after intranasal infection.**
Rabbits were infected by intranasal inoculation of different doses ($5{\times}10^4$, $2{\times}10^5$ or $8{\times}10^5$ PFU per rabbit) of C500 BAC⁻ WT and BAC⁻ A7$^{STOP-207}$. (A) Percentages of CD8+ T cells in PBMCs analyzed by flow cytometry at regular intervals throughout the experiment. Data are plotted for individual rabbits. (B) Percentages of CD8+ T cells in PBMCs, popliteal lymph nodes (pLN) and spleen at the time of euthanasia, as analyzed by flow cytometry. Data show mean ± SEM with data plotted for each individual rabbit. (C) qPCR of viral genome copies in PBMCs over time. (D) qPCR of viral genome copies in PBMCs, pLN and spleen at the time of euthanasia. Real-time PCR quantification was normalized on $10^5$ copies of the cellular β-globin gene sequence. Data are plotted as individual measurements. Bars show mean values. One-way ANOVA with Bonferroni's post-hoc test was used to identify significant differences (***$P{<}0.001$).

We and others have previously described the presence of genome rearrangements in C500 and viruses derived from the C500 (WT) BAC clone after propagation in vitro [36,49,50]. Thus, AlHV-1 attenuation is often associated with the selection of more efficient growth and cell-free transmission in vitro. The current study involved two bovine cell lines of respiratory origin, BT and EBL cells, to investigate the spread of WC11 and C500 in vitro in the presence or absence of A7 or A8. These experiments demonstrated that WC11 propagates largely in a cell-free manner, reaching high titers in comparison to C500. Moreover, BT cells were identified as being permissive to cell-fee propagation, whereas EBL cells largely propagated infectivity in a

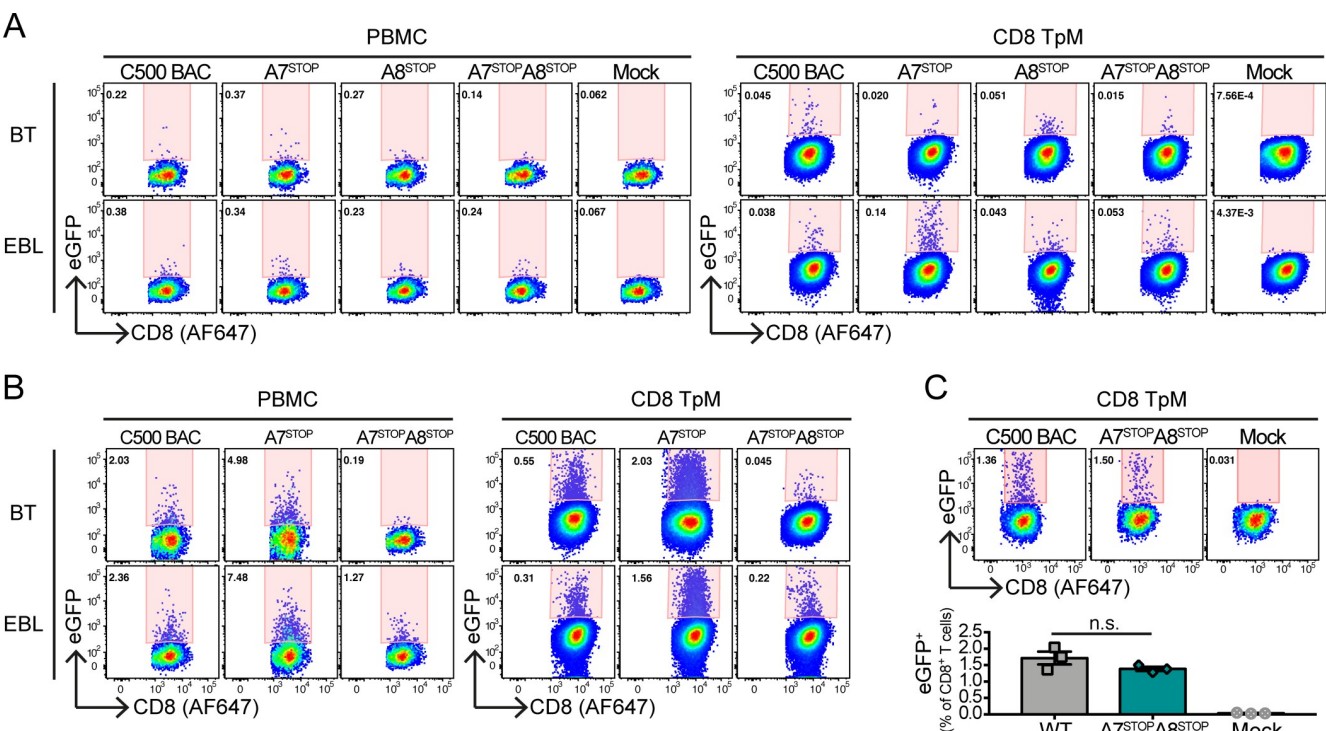

**Fig 10. Ex vivo infection of bovine CD8+ T lymphocytes by co-culture with infected BT or EBL cell.** (A and B) Pseudocolour contour plots showing the percentages of eGFP+ cells in gated CD8+ T lymphocytes (PBMCs or CD8TpM cell line) by flow cytometry after 48h co-culture with BT or EBL infected with C500 BAC+ WT, BAC+ A7STOP, BAC+ A8STOP or BAC+ A7STOPA8STOP viruses (A = m.o.i. = 0.01; B = m.o.i. = 0.1). (C) Representative pseudocolour contour plots of CD8TpM cells showing the percentages of eGFP+ cells after co-culture with EBL infected with C500 BAC+ WT or BAC+ A7STOPA8STOP virus. Bars show mean ±SD (n = 3). One-way ANOVA with Dunnet's post-hoc test was used to identify significant differences (n.s. = not significant).

cell-associated manner. Using both cell lines, it appeared that A7 expression is essential for cell-associated propagation and syncytia formation, whereas A8 is involved in cell-free propagation in BT cells. These results were supported by the use of A7STOP, A8STOP and double A7STOPA8STOP mutant viruses. The lack of A7 expression resulted in increased viral growth in BT cells with increased levels of cell-free virions. Thus, although the lack of A7 generated very small plaques and impaired infectivity in EBL cells, viral propagation and entry were significantly enhanced in BT cells. Conversely, the absence of A8 had little effect on viral propagation and syncytia formation in EBL cells, but resulted in impaired viral growth in BT cells. Interestingly, double A7STOPA8STOP mutant virus was not able to reach growth kinetics similar to WC11 or even A7STOP in BT cells whereas it displayed a similar phenotype to the latter in EBL cells. These results illustrate the importance of both proteins in regulating viral propagation and may explain, while not completely, why WC11 lost both A7 and A8 while being selected for propagation in vitro. It is possible that A1 and microRNAs present in the left part of the genome or sequence divergence in A9.5 and 10 would explain the further enhanced fitness of WC11. Moreover, there remains a truncated part of A8 in WC11 that might still have a role in the attenuated strain. These hypotheses need to be addressed in future work. The results presented in this study also suggest that the A7 protein may mediate cell-to-cell viral spread through the promotion of membrane fusion and the retention of virions at the infected cell surface, whereas the A8 protein may trigger attachment of cell-fee virions to target cells in order to facilitate viral entry and propagation.

Among the few major differences between the WC11 and C500 sequences, the lack of A7 or A8 expression may explain the inability of WC11 to induce MCF. Interestingly, the roles of gp42 or gp350 in the regulation of EBV propagation and tropism in vitro have been investigated extensively [26], but their roles in the induction of EBV-induced lymphoproliferative disorders in vivo have not been investigated directly owing to the lack of a suitable animal model. Whereas gp350 orthologs are encoded in the majority of gammaherpesviruses, gp42 orthologs are present only in the genera *Lymphocryptovirus*, *Percavirus* and *Macavirus*. Infection of rabbits with C500 mutants unable to express A7 or A8 resulted in the absence of MCF development, with no hyperthermia and the absence of lymphadenopathy or splenomegaly, CD8$^+$ T lymphocyte expansion and detectable viral genomes in lymphoid tissues. Although rabbits are not naturally susceptible to AlHV-1 infection, they develop clinical signs and lesions typical of MCF upon experimental infection and have been used extensively as a model for investigating the pathogenesis of the disease [17–19,51]. Thus, infection of rabbits demonstrated that expression of both A7 and A8 is necessary to induce MCF lesions, including expansion of latently infected CD8$^+$ T cells. While no or very low copy numbers of AlHV-1 were detected in peripheral blood and lymph nodes after infection with A7$^{STOP}$, A8$^{STOP}$ or WC11, low level of viral genomes were still detectable in spleen, lung and liver tissues. Rabbits infected with pathogenic strain C500 showed characteristic infiltrates of lymphoblastic cells in the lung and liver, which likely contain a high proportion of latently infected CD8$^+$ T cells [17,18]. However, rabbits infected with the A7$^{STOP}$, A8$^{STOP}$ or WC11 viruses did not show MCF lesions but virus genomes were still detected. Thus, absence of A7 and/or A8 might affect viral spread to target CD8$^+$ T cells in vivo while does not prevent AlHV-1 ability to persist in infected animals. The cellular niche of such persistence still needs to be uncovered. The presence of AlHV-1-specific antibodies, including neutralizing antibodies, further supports that all rabbits were effectively infected. Moreover, higher titers of neutralizing antibodies were observed in rabbits infected with A7$^{STOP}$, which suggests that increased viral replication may have occurred at primary infection sites in vivo or that virions devoid of A7 may be more efficient at inducing neutralizing antibodies.

Based on our findings, it appears that A7 and/or A8 expression are necessary for AlHV-1 to induce MCF by regulating viral spread in vivo but we could not provide any functional evidence that A7 and/or A8 would be involved in the direct infection of CD8$^+$ T lymphocytes. Indeed, in absence of A7 and/or A8, AlHV-1 could effectively infect bovine CD8$^+$ T cells obtained from PBMCs or a CD8$^+$ T cell line immortalized with the protozoa *T. parva*. These results were important as they provide a novel mechanistic insight on the role of those viral glycoproteins in regulating viral spread in vivo and reach target CD8$^+$ T cells through an A7/A8-independent mechanism and induce their malignant proliferation. Although A7 and/or A8 could be involved in reprogramming CD8$^+$ T lymphocytes upon infection, such hypothesis is highly unlikely as RNA expression of these genes could not be detected in lymph nodes of calves developing MCF [18].

We have shown that A7 and A8 encode orthologs of EBV gp42 and gp350, respectively, and that they tightly regulated cell-associated viral spread and cell-free viral propagation in two bovine cell lines of respiratory origin and were essential for inducing MCF. These results highlight the importance of targeting early events in gammaherpesvirus infection as a mean of intervention. Indeed, whereas the leading EBV vaccine antigen gp350 does not protect B cells from infection, recent development of a nanoparticle vaccine strategy involving the gH/gL/gp42 viral fusion machinery resulted in complete neutralization of epithelial cell and B cell infection by EBV [52]. Immunization of rabbits with a live, non-pathogenic, ORF73-deleted AlHV-1 strain has been shown to result in complete protection against a virulent challenge with AlHV-1 [18]. Development of inactivated vaccine strategies targeting the viral entry

machinery and including A7 or A8 to prevent MCF development would open new avenues for cost-effective intervention against AlHV-1 in endangered species and livestock in regions where infection is endemic.

## Materials and methods

### Ethics statement

The experiments, maintenance and care of rabbits complied with the guidelines of the European Convention for the Protection of Vertebrate Animals used for Experimental and other Scientific Purposes (CETS n° 123). The protocol was approved by the Committee on the Ethics of Animal Experiments of the University of Liège, Belgium (Permits #1127, 1571). All efforts were made to minimize suffering.

### Homology modeling of predicted protein structures

The three-dimensional structures of the A7 and A8 proteins were predicted on the basis of the crystal structures of EBV gp42 [53] and gp350 [47], respectively. The structures of gp42 (3FD4. PDB) and gp350 (2H6O.PDB) were obtained from the Protein Data Bank (https://www.rcsb. org) and used as models to predict the putative folding of the extracellular domains of A7 and A8 using the protein structure and function predictions server I-TASSER (http://zhanglab. ccmb.med.umich.edu/I-TASSER/). Alignments of the resulting models displaying the highest C-score were performed using PyMol 1.3. Sequence alignments were obtained using CLC Sequence Viewer v.7.8.1.

### Cell lines and viruses

Bovine turbinate (BT) fibroblasts (American Type Culture Collection (ATCC) CRL-1390), embryonic bovine lung (EBL) cells (German Collection of Microorganisms and Cell Cultures (DSMZ) ACC192), MacT cells [54] and MacT-Cre cells [36] were cultured in Dulbecco's modified essential medium (DMEM, Invitrogen Corporation). Madin-Darby bovine kidney (MDBK) cells (ATCC CCL-22) were cultured in minimum essential medium (MEM). All cell lines were cultured in presence of 10% (v/v) fetal calf serum (FCS, BioWhittaker). AlHV-1 strains C500 (generated from the BAC clone; C500 (WT)) and WC11 were used in this study (strains C500 and WC11 were originally obtained from Prof. David M. Haig, University of Nottingham, UK). C500 BAC virus (WT) and derived mutants were maintained by limited passage (<5) in BT cells [44]. *T. parva*-immortalized CD8$^+$ T cell clone 641 (CD3$^+$CD4$^-$CD8$^+$NKp46$^-$γδTCR$^-$) was maintained in RPMI-1640 containing 10% (v/v) FCS, 2-mercaptoethanol (2 mM) and gentamicin (50 μg/mL).

### Production of A7 and A8 mutants

The C500 BAC WT clone was used to generate A7 and A8 nonsense mutants using the galactokinase gene (galk) as a selectable marker in *E. coli* by recombineering [55]. To produce the A7$^{STOP}$ BAC clones, a stop codon was introduced at the 5' end of the A7 coding region. The A7$^{STOP-39}$ BAC clone was produced by inserting a mutation resulting in an in-frame stop codon at position 49–51 (GenBank accession AF005370, nt 76618–76620). Because the 5'-end of the A7 sequence is present in a duplicated region, and to minimize the risk of reversion, a A7$^{STOP-207}$ mutant with an in-frame stop codon at position 207–209 (GenBank accession AF005370, nt 76777–76779) was also generated. The galK gene was introduced via a recombination fragment generated by PCR using the pgalK plasmid as template and chimeric primers A7-NS-galkFwd39 or A7-NS-galkFwd207 and A7-NS-galkRev39 or A7-NS-galkRev307 (S3

Table). Then, oligonucleotides A7-NS-oligoFwd39 or A7-NS-galkFwd207 and A7-NS-oligoRev39 or A7-NS-galkRev207 were annealed to introduce the mutations and an EcoRI restriction site by galk-dependent negative selection. A similar strategy was employed to produce the A8$^{STOP}$ BAC, using primers A8-NS-galkFwd and A8-NS-galkRev to insert galK into the target site. Then, oligonucleotides A8-NS-oligoFwd and A8-NS-oligoRev were annealed in order to insert a mutation and an EcoRI restriction site. To generate A7$^{STOP}$A8$^{STOP}$ double mutants, an identical strategy was used to insert STOP codons in A7 coding sequence in the A8$^{STOP-159}$ BAC clone. All strains were reconstituted into viruses by transfection into MacT (to generate BAC$^+$ viruses) and MacT-cre (to generate BAC$^-$ viruses) cells before propagating in BT cells. BAC$^+$ viruses were used in experiments in vitro to use eGFP as a reporter protein to monitor viral infection and BAC$^-$ viruses were used in experiments in vivo.

## Southern blotting

DNA preparations were digested with EcoRI, separated on a 0.7% agarose gel, and transferred to a Amersham Hybond-XL membrane (GE Healthcare) by capillary transfer as described previously [17]. DNA fragments containing A7 or A8 were detected using hybridization probes generated by PCR using primers A7-165F and A7-303R or A8-ATGF and A8-278R, respectively. Probes were generated with α-[$^{32}$P] dCTP (specific activity, 3000 Ci/mmol; Perkin Elmer) using the random-primed DNA labeling kit (Roche).

## DNA sequencing

WC11 and viruses generated from C500 BAC clones were grown in BT cells and purified from the medium by ultracentrifugation (100,000×$g$) through a 30% (w/v) sucrose cushion for 2h at 4˚C. DNA was extracted from the pelleted virus [44]. BAC clone DNA was purified using the Large Construct DNA prep kit (Qiagen). Paired-ended sequence reads of 250 nt (WC11) or 150 nt (C500 BAC mutants and viruses derived therefrom) were generated using a MiSeq DNA sequencer running version 2 chemistry (Illumina) and assembled using methods described previously [36] (S2 Table).

## Nucleotide sequence accession numbers

The sequence and annotation of the full sequence of the strain WC11 was deposited in GenBank under accession number KX905136.2.

## cDNA synthesis and real-time PCR

Total RNA was extracted from infected cells using a RNeasy miniprep kit (Qiagen) and treated with DNAse I using the TURBO DNA-free kit (Ambion) to remove remaining DNA. Then, 300 ng RNA was used to synthesize cDNA using the iScript cDNA synthesis kit (Bio-Rad Laboratories). Reactions lacking reverse transcriptase were carried out in order to confirm the absence of viral DNA in subsequent PCR reactions. Quantitative PCR reactions were performed using primer pairs A7-476F/deltastopR and A8-1661F/deltastopR (S3 Table) and iQ SYBR Green Supermix (Bio-Rad) on a CFX96 Touch Real-Time PCR Detection System (Bio-Rad).

## Multistep growth curves.

Cells were infected (m.o.i. = 0.01 PFU/cell), and medium and cells were harvested at successive intervals. The total amount of infectious viral particles (PFU) was determined by plaque assay as described previously [18].

## Plaque size assay.

Cells grown on glass coverslips were infected and overlaid after 3h with DMEM containing 10% (v/v) FCS and 0.6% (w/v) carboxymethylcellulose (CMC; Sigma). Plaque areas were then fixed in PBS containing 4% paraformaldehyde (PFA) before being permeabilized in PBS containing 0.1% Nonidet P-40. Then, monolayers were stained using anti-AlHV-1 rabbit polyserum as primary antibody (1:500 dilution) followed by Alexa Fluor 488-conjugated polyclonal goat anti-rabbit IgG secondary antibody (Thermo Fischer, 2 μg/mL). In some experiments, mouse monoclonal antibody IgG2b clone 15-A was used as primary antibody (2 μg/mL) followed by Alexa Fluor 568-conjugated polyclonal goat anti-rabbit IgG secondary antibody (Thermo Fischer, 2 μg/mL). Microphotographs were taken using an epifluorescence microscope using the system described below. Areas were then measured using ImageJ v1.46q as described previously [8].

## Infectivity assay

BT or EBL cells ($2 \times 10^5$ cells/well) were inoculated with BAC$^+$ viruses (m.o.i. = 0.05 PFU/cell) before washing three times with cold phosphate-buffered saline (PBS) at different time-points after inoculation. At 20 h post-infection, the cells were trypsinized and single-cell suspensions analyzed by flow cytometry to obtain the percentage of eGFP+ cells. Dead cells were excluded using 7-aminoactynomycin D (7-AAD, Sigma Aldrich).

## Infection of rabbits

Intravenous inoculations were performed by administering the equivalent of 50 cm$^2$ of infected BT cells displaying >90% cytopathic effect per rabbit, as described previously [17]. To measure the infectivity of the live cell inoculum administered intravenously to the rabbits, infectious center assays (ICA) was performed as previously described [17]. IC numbers were obtained for each inoculum (WT: $2.0 \times 10^4$, A8$^{STOP}$: $2.1 \times 10^4$, A7$^{STOP}$: $1.1 \times 10^5$, WC11: $0.9 \times 10^5$ ICs/rabbit). Intranasal inoculations of different doses ($5 \times 10^4$, $2 \times 10^5$, $8 \times 10^5$ PFU per rabbit) were performed by administering concentrated virus via instillation in 0.5 mL DMEM into the nostrils of anesthetized rabbits [23]. Rabbits were monitored daily for clinical signs. In compliance with bioethical guidance, rabbits were euthanized when rectal temperature remained higher than 40°C for 2 days consecutively.

## Leukocyte cell suspension preparation

Rabbit PBMCs were isolated from 5 mL blood collected from the central artery of the rabbit ear before infection and at various time-points post-infection. Immediately after euthanasia, single-cell suspensions were prepared from popliteal lymph nodes and spleen as follows. Tissues were gently disrupted with scissors in sterile RPMI-1640 medium and passed through a 70 μm cell-strainer using a 1-mL syringe plunger. Mononuclear leukocyte suspensions were prepared from peripheral blood and tissue samples using Ficoll-Paque Premium density gradient medium (GE Healthcare). Each 5 mL single-cell suspension was diluted 1:1 in sterile PBS, overlaid onto a 5 mL Ficoll-Paque density cushion, and centrifuged (1825×*g*) for 20 min at room temperature. Mononuclear leukocytes at the interface were collected and washed in ice-cold PBS before further analysis. Bovine blood was obtained from a healthy donor of CARE--FEPEX ("Cellule d'Appui à la Recherche et à l'Enseignement–Ferme Pédagogique et Experimentale", Dr L. Martinelle) and PBMCs isolated on Ficoll-Paque density gradient using SepMate isolation tubes (STEMCELL technologies).

## Quantification of viral genomes

Total DNA was extracted from single-cell suspensions in PBS (PBMCs and popliteal lymph nodes) or tissue biopsies (spleen, liver and lung) using the QiaAmp DNA mini kit (Qiagen). Before DNA extraction, tissues were homogenized using a TissueLyzer II (Qiagen). Then, AlHV-1 DNA copies were quantified using an AlHV-1 ORF3 real-time PCR and normalized to the cellular beta-globin sequence, as described previously [10], using iQ Supermix (Bio-Rad) and a CFX96 Touch Real-Time PCR Detection System (Bio-Rad).

## Antibodies and flow cytometry

Multi-colour flow cytometry analysis was performed as described previously [19]. Briefly, cells were incubated with a cocktail of monoclonal antibodies against rabbit CD4 (IgG2a, KEN-4), CD8 (IgG1, 12C.7) and IgM (IgG1, NRBM) on ice for 10 min. Cells were washed and further incubated for 10 min on ice with isotype-specific phycoerythrin (PE)-conjugated rat anti-mouse IgG1 (A85-1, BD) and biotinylated rat anti-mouse IgG2a (R19-15, BD) antibodies. After a third wash, cells were incubated with fluorescein isothiocyanate (FITC)-conjugated anti-rabbit T cells (KEN-5), washed with allophycocyanin (APC)-conjugated streptavidin (BD), and suspended in 7-AAD. Antibodies were from AbD-Serotec. Bovine CD8$^+$ T cells were identified as previously described [18], and dead cells detected with BD Horizon Fixable Viability Stain 450 (BD biosciences). Data were acquired using a Fortessa X20 flow cytometer (BD) and analyzed using Flowjo v10.0.7 (Treestar).

## Seroneutralization

Serum dilutions were incubated with C500 BAC$^+$ virus (100 PFU/50 uL serum dilution) for 2h at 37˚C. MDBK cells were then incubated with the neutralized inoculum and overlaid after 3 h with DMEM containing 10% (v/v) FCS and 0.6% (w/v) CMC.

## Histological analysis

Organ explants from mock-infected or infected rabbits were fixed in 10% (v/v) buffered formalin and embedded in paraffin blocks. Five-micron sections were then stained with hematoxylin and eosin prior to microscopic analysis.

## Microscopy

Microscopic analyses were performed using epifluorescence DM2000 LED microscope (Leica) equipped with a DF450 C camera (Leica) or an Eclipse TE2000-S inverted microscope (Nikon) equipped with a DC 300F camera (Leica). Images were acquired using the Leica Application Suite 4.2 software (Leica).

## Statistical analyses

Statistical analyses were conducted using GraphPad Prism v6.

## Supporting information

**S1 Fig. Glycosylation prediction.** Glycosylation sites were predicted using NetNGlyc 1.0 (http://www.cbs.dtu.dk/services/NetNGlyc/) and NetOGlyc 4.0 (http://www.cbs.dtu.dk/services/NetOGlyc/) glycosylation site predictors.
(TIF)

**S2 Fig. Sequence analysis of the AlHV-1 strain C500 A7 and A8 proteins.** (A) Clustal Omega was used to align the primary amino acid sequences of the A7 protein and EBV gp42. The C-type lectin-like domain is highlighted in yellow (InterPro Protein sequence analysis & classification - https://www.ebi.ac.uk/interpro/). Conserved residues are shown in red. (B) Prediction of the three-dimensional structure of the extracellular domain of the A7 protein based on the structure of EBV gp42 [53]. (C) Clustal Omega was used to align the primary amino acid sequences of the A8 protein and EBV gp350. The extracellular domain is highlighted in yellow. (D) Prediction of the three-dimensional structure of the extracellular domain of the A8 protein based on the structure of EBV gp350 [47]. Transmembrane domains are highlighted in cyan (TMHMM Server v. 2.0), and conserved residues are shown in red.
(TIF)

**S3 Fig. Production of AlHV-1 mutants impaired for A7 or A8 expression.** (A) The C500 BAC clone (WT) was used to produce the A7$^{STOP-39}$, A7$^{STOP-207}$, A8$^{STOP-159}$, A7$^{STOP-39}$A8$^{STOP-159}$ and A7$^{STOP-207}$A8$^{STOP-159}$ BAC clones by mutagenesis. Galactokinase (galK)-based recombineering methodology for positive and negative selection was used to introduce an in-frame stop codon into the A7 or A8 coding sequence followed by an EcoRI restriction site (in italics). The stop codons generated are in bold font and underlined in red. A7$^{STOP-39}$A8$^{STOP-159}$ and A7$^{STOP-207}$A8$^{STOP-159}$ BAC clones were produced from A8$^{STOP-159}$ BAC clone. (B) Southern blotting of the BAC clones generated. The A7 and A8 probes were produced by PCR as described in Methods. EtBr indicates ethidium bromide-stained lanes prior to blotting. The entire gel and Southern blots are displayed. (C) Sequence alignments of the mutagenesis performed to generate the mutants and the sequencing data obtained for BAC clones (Plasmid) and the BAC⁻ viruses (Virus) generated from them. The first nucleotide of start codons is shown in bold black font and the stop codons are shown in bold red font. The positions of frameshifts in amino acid sequences are shown by bold red asterisks.
(TIF)

**S4 Fig. Monitoring of body temperature over time.** (A) Rabbits were infected by intravenous inoculation with 50 cm$^2$ of mock-infected BT cells or cells infected with C500 BAC WT, A7$^{STOP-207}$, A8$^{STOP-159}$ or WC11 virus. (B) Rabbits were infected by intranasal inoculation of different doses ($5\times10^4$, $2\times10^5$ or $8\times10^5$ PFU per rabbit) of C500 BAC WT and A7$^{STOP-207}$ virus.
(TIF)

**S5 Fig. Neutralizing antibody response.** (A) Rabbits were infected by intravenous inoculation with 50 cm$^2$ of mock-infected BT cells or cells infected with C500 BAC WT, A7$^{STOP-207}$, A8$^{STOP-159}$ or WC11 virus. (B) Rabbits were infected by intranasal inoculation of $5\times10^4$ PFU per rabbit of C500 BAC WT and A7$^{STOP-207}$virus. Serum samples were tested for neutralizing antibodies at day 7, 14 and 21.
(TIF)

**S6 Fig. Gating strategy for CD8$^+$ T cells (PBMCs or CD8TpM) after co-culture with BT or EBL cells.**
(TIF)

**S1 Table. AlHV-1 strain WC11 genome organization compared to other Macaviruses.**
(PDF)

**S2 Table. Whole AlHV-1 genome sequencing data.**
(PDF)

**S3 Table. Oligonucleotides.**
(PDF)

## Acknowledgments

B.G.D. is a Research Associate of the "Fonds de la Recherche Scientifique" (F.R.S.-FNRS). The authors are thankful to Lola Dechêne, Lorène Dams, Cédric Delforge, Aurélie Vanderlinden and Emeline Deglaire for excellent technical assistance.

## Author Contributions

**Conceptualization:** Françoise Myster, Benjamin G. Dewals.

**Data curation:** Françoise Myster, Mei-Jiao Gong, Benjamin G. Dewals.

**Formal analysis:** Françoise Myster.

**Funding acquisition:** Andrew J. Davison, Benjamin G. Dewals.

**Methodology:** Françoise Myster, Justine Javaux, Nicolás M. Suárez, Gavin S. Wilkie, Tim Connelley, Benjamin G. Dewals.

**Supervision:** Benjamin G. Dewals.

**Writing – original draft:** Françoise Myster, Benjamin G. Dewals.

**Writing – review & editing:** Françoise Myster, Alain Vanderplasschen, Andrew J. Davison, Benjamin G. Dewals.

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
