## [Decision Letter · Decision Letter 0]

31 Oct 2019

Dear Prof. Dewals,

Thank you very much for submitting your manuscript "Alcelaphine herpesvirus 1 genes A7 and A8 regulate viral spread and are essential for malignant catarrhal fever" (PPATHOGENS-D-19-01549) for review by PLOS Pathogens. Your manuscript was fully evaluated at the editorial level and by independent peer reviewers. The reviewers appreciated the attention to an important problem, but raised some substantial concerns about the manuscript as it currently stands. These issues must be addressed before we would be willing to consider a revised version of your study. We cannot, of course, promise publication at that time.

I am returning your manuscript with two reviews. The reviewers came to different conclusions about the paper, as you will see. After reading the reviews and looking at the manuscript, I recommend Major Revision based on the critiques from the more critical reviews. I am sorry I cannot be more positive at the moment, however we are looking forward to receiving your revision. With a lot of work, the manuscript will be suitable for a resubmission, if you so wish to do so.

Please pay particular attention to the following reviewer suggestions and give them due consideration:

-Comments related to generation of a double mutant and the design of the rabbit studies to distinguish between access versus reprogramming of T cells

-provide data for infection of bovine and rabbit T cells as described by reviewer 2

Your revisions should address the specific major and minor points made by each reviewer.

(1) A letter containing a detailed list of your responses to the review comments and a description of the changes you have made in the manuscript. Please note while forming your response, if your article is accepted, you may have the opportunity to make the peer review history publicly available. The record will include editor decision letters (with reviews) and your responses to reviewer comments. If eligible, we will contact you to opt in or out.

(2) Two versions of the manuscript: one with either highlights or tracked changes denoting where the text has been changed; the other a clean version (uploaded as the manuscript file).

Additionally, to enhance the reproducibility of your results, PLOS recommends that you deposit your laboratory protocols in protocols.io, where a protocol can be assigned its own identifier (DOI) such that it can be cited independently in the future. For instructions see http://journals.plos.org/plospathogens/s/submission-guidelines#loc-materials-and-methods

We hope to receive your revised manuscript within 60 days. If you anticipate any delay in its return, we ask that you let us know the expected resubmission date by replying to this email. Revised manuscripts received beyond 60 days may require evaluation and peer review similar to that applied to newly submitted manuscripts.

[LINK]

Sincerely,

Paul D. Ling, Ph.D.

Associate Editor

PLOS Pathogens

Erik Flemington

Section Editor

PLOS Pathogens

Kasturi Haldar

Editor-in-Chief

PLOS Pathogens

orcid.org/0000-0001-5065-158X

Grant McFadden

Editor-in-Chief

PLOS Pathogens

orcid.org/0000-0002-2556-3526

Reviewer's Responses to Questions

**Part I - Summary**

Reviewer #1: This paper describes an analysis of the attenuated AlHV1 strain WC11 and of the production and phenotypes of two BAC clone with stop mutations in genes deleted in WC11. The study is an important step forward in understanding the basis of attenuation in MCF viruses. It provides a well written and thorough analysis of the A7 and A8 genes, showing that these proteins mediate cell-free and cell-associated modes of propagation and that loss of either function leads to loss of virulence for MCF.

The paper is suitable for publication in its current state but may benefit from minor modifications as described in the comments for authors.

Reviewer #2: Myster and coauthors address the genetic differences between the pathogenic C500 and an attenuated WC11 strain of the Macavirus Alcelaphine herpesvirus 1 (AlHV-1). AlHV-1 and Ovine Herpesvirus 2 induces malignant catarrhal fever in susceptible ruminants and in rabbits. In cattle and rabbits, MCF is characterized by proliferation of CD8+ T-cells, a lymphoproliferative disease with strong similarity to the pathology caused by the New World primate viruses herpesvirus saimiri and ateles in tamarins and marmosets (which is overlooked in the introduction). They focus on one of three major differences between the pathogenic and the attenuated strains, the putative glycoproteins encoded by ORFs A7 and A8, which are both lacking in the attenuated WC11 strain. Individual bacmid-technology based mutants of each of A7 and A8 were generated in the C500 strain and compared to C500 and the A7 and A8 deleted WC11 in vitro and in vivo. Although no marker rescue viruses were made, the recombinants were sequenced and the data obtained with the mutants seems consistent. However, the important rabbit experiments do not discriminate between an inability to spread efficiently in the rabbits, failure to gain access to CD8 T cell populations or an inability to reprogram such cells toward lymphoproliferation.

The authors provide an extensive comparison of the replicative behavior of C500, WC11 and two A7, one A8 mutants and mutants. Herein, depending on the cell culture system, the A7 mutants show an intermediate phenotype, increasing cell free propagation, while A8 seems to be required for cell associated spread. This may e.g. reflect alternative receptor usage for entry in these culture systems.

Finally, rabbit experiments show that the C500 A7stop, A8stop and, as expected, attenuated WC11 do not induce MCF or CD8 T cell lymphoproliferation in this experimental model. They provide evidence all strains were able to infect rabbits, by serology and detection of viral genomes. The mutants and attenuated WC11, however, were only detectable at very low levels in the spleen, and not in circulating PBMC or lymph node.

**Part II – Major Issues: Key Experiments Required for Acceptance**

Reviewer #1: (No Response)

Reviewer #2: Critical points

- Why did they not make / analyze a double A7 and A8 stop mutant, corresponding to the situation in WC11? If the double mutant would show a replicative behavior even more similar to WC11, this would be important with respect to possible influence of the other two major genetic differences, which are located on the left and right genome ends, i.e. the absent putative ORF A1 and the variable ORFs A9.5 and A10?

- The rabbit experiments do not discriminate between an inability to spread efficiently in the rabbits, failure to gain access to CD8 T cell populations or an inability to reprogram such cells toward lymphoproliferation. There was no monitoring for viral spread (genome copies in blood?) during the infection experiment, only at endpoint. Thereby, vastly different time points are compared, and the low levels of virus genomes observed with A7 and A8 stop mutants and WC11 may reflect long term immune control of latent infection, compared to an acute disease with multiple virus genomes in the CD8 population.

- In vitro infection (or by cocultivation) of bovine or rabbit CD8 T-cells with the EGFP marker viruses (or by infection followed by PCR) could demonstrate whether this is a matter of entry/access to this population.

- Cells injected intravenously, if not phagocytosed or killed the immune cells, may end in small vessels of lung, spleen, liver etc. where they may infect neighboring tissue, not necessarily T cells. Similarly, after intranasal infection, local infection and spread is critical, and T cells are unlikely to be the first cells encountering a pathogen.

**Part III – Minor Issues: Editorial and Data Presentation Modifications**

Reviewer #1: Minor issues

Line 150. Please be explicit. Clarify whether the LUR or TR copies are meant

Line 153. Please specify German WC11 stock to distinguish from the stock sequenced here

Line 157-8. please clarify text here, as this point may be unclear to some readers

Line 160. "during multiple passages" rather than ‘following multiple passages’

Lines 194-209. the authors should acknowledge previous publications describing the similarity between A7/A8 and gp42/350 (Russell, 2014. In: Manual of Security Sensitive Microbes and Toxins. CRC Press; https://www.crcpress.com/Manual-of-Security-Sensitive-Microbes-and-Toxins/Liu/p/book/9780367378745); and the function of the A8 homolog Ov8 in membrane fusion (AlHajri, 2017. J.Virol. 91:e02454-16. https://doi.org/10.1128/JVI.02454-16).

Line 217. In the BAC mutation description it is important that you show that both TR and LUR gene copies of A7 & A8 were mutated, or explain why this was not required

Line 251. Fluorescence microscopy and use of antibody 15A are not mentioned in methods

Line 300. is ref 43 intended? Maybe should be 35?

Line 302. While it’s completely acceptable to choose to study A7/A8, I think it is important that the authors mention why there was no focus on A1 in this work.

Line 342. I don’t feel that sup fig 4 is necessary

Reviewer #2: Minor points:

- Figs 1,3,4 are more suitable for supplement.

- For intravenous injection, infected tissue culture cells are used, while the authors refer to pfu for intranasal infection?

PLOS authors have the option to publish the peer review history of their article (what does this mean?). If published, this will include your full peer review and any attached files.

Reviewer #1: Yes: George C. Russell

Reviewer #2: No

---

## [Editor Report · Decision Letter 1]

17 Feb 2020

Dear Prof. Dewals,

We are pleased to inform you that your manuscript 'Alcelaphine herpesvirus 1 genes A7 and A8 regulate viral spread and are essential for malignant catarrhal fever' has been provisionally accepted for publication in PLOS Pathogens.

Before your manuscript can be formally accepted you will need to complete some formatting changes, which you will receive in a follow up email. A member of our team will be in touch within two working days with a set of requests.

Best regards,

Paul D. Ling, Ph.D.

Associate Editor

PLOS Pathogens

Erik Flemington

Section Editor

PLOS Pathogens

Kasturi Haldar

Editor-in-Chief

PLOS Pathogens

orcid.org/0000-0001-5065-158X

Michael Malim

Editor-in-Chief

PLOS Pathogens

orcid.org/0000-0002-7699-2064
---

## [Editor Report · Acceptance letter]

10 Mar 2020

Dear Prof. Dewals,

We are delighted to inform you that your manuscript, "Alcelaphine herpesvirus 1 genes A7 and A8 regulate viral spread and are essential for malignant catarrhal fever," has been formally accepted for publication in PLOS Pathogens.

Best regards,

Kasturi Haldar

Editor-in-Chief

PLOS Pathogens

orcid.org/0000-0001-5065-158X

Michael Malim

Editor-in-Chief

PLOS Pathogens

orcid.org/0000-0002-7699-2064